# Association between Food Label Unawareness and Loss of Renal Function in Diabetes: A Cross-Sectional Study in South Korea

**DOI:** 10.3390/ijerph17061945

**Published:** 2020-03-16

**Authors:** Jae Hong Joo, Doo Woong Lee, Dong-Woo Choi, Eun-Cheol Park

**Affiliations:** 1Department of Public Health, Graduate School, Yonsei University, Seoul 03722, Korea; jhj3040@yuhs.ac (J.H.J.); doowoonglee@yuhs.ac (D.W.L.); CDW6027@yuhs.ac (D.-W.C.); 2Institute of Health Services Research, Yonsei University, Seoul 03722, Korea; 3Department of Preventive Medicine, Yonsei University College of Medicine, Seoul 03722, Korea

**Keywords:** diabetic patients, renal function, food label unawareness, estimated glomerular filtration rate

## Abstract

Objectives: To examine sex differences in the association between food label unawareness and loss of renal function among South Korean diabetic patients aged ≥30 year and determine whether reading food labels when choosing which food products to consume plays a potential role in slowing the progression of renal disease. Methods: Data from the 2016–2017 Korea National Health and Nutrition Examination Survey were used for the analysis. Renal function was determined by the Modification of Diet in Renal Disease estimated glomerular filtration rate, and food label unawareness was defined as being unaware of the food label when choosing a food product for consumption. Multiple regression analysis was used to investigate the association between food label unawareness and loss of renal function among South Korean diabetic patients. Results: Four hundred and eighty-seven diabetic patients (men: 274; women: 213) were enrolled. Loss of renal function was associated with food label unawareness in only male diabetic patients (men: *β* = –10.01, standard error (SE) = 5.08, *p* = 0.0506; women: *β* = –0.30, SE = 5.14, *p* = 0.9528). A strong association was found between loss of renal function and food label unawareness among socially isolated male diabetic patients who lived in a one-generational household, did not have a spouse, and ate alone. Conclusion: Cultivating habits of reading food labels and inducing social facilitation may play a potential role in managing loss of renal function among male diabetic patients.

## 1. Introduction

Renal disease is a global public health concern, and its prevalence has been gradually increasing in conjunction with an increase in the incidence of diabetes mellitus [1,2]. There has been convincing evidence that adoption of a Westernized lifestyle contributes to the increased diagnosis of diabetes mellitus in Asia [3]. It has been reported that about 4.8 million Koreans (13.7%) had diabetes in 2014, and nearly one-third of people with diabetes had albuminuria or decreased renal function [4]. Untreated diabetes is the most common cause of end-stage renal disease and other organ complications, which contributes to the increasing mortality rate [5,6]. Diabetes can potentially be reversed, yet many people tend to live with it because it develops at an old age and is a serious long-term condition followed by expensive clinical effort for adequate treatment [7]. Through pertinent management, the progression of renal disease can be delayed, thus easing the burden of diabetic patients [8].

Practicing a healthy dietary habit is key and probably one of the most cost-effective methods for attenuating the morbidity and mortality associated with chronic diseases [9,10]. The use of food labels is associated with nutrition knowledge [11]. The awareness of nutritional facts labeled on food product packages/menus can improve one’s dietary intake patterns [12]. Given that dietary intake patterns are inversely related with obesity, it is important for individuals to make healthier dietary choices and keep track of what they consume. Excessive weight gain induces renal sodium reabsorption by activating the renin–angiotensin and sympathetic nervous systems, and this alters renal structure [13]. Furthermore, sustained structural changes in the kidney causes loss of nephron function, which further increases arterial pressure and leads to severe inflammation in the renal system [14].

The purpose of this study was to examine the association between food label unawareness and the loss of renal function in South Korean diabetic patients, and determine whether reading food labels when choosing which food products to consume plays a potential role in slowing the progression of renal disease. This study hypothesized that not using food labels is associated with decreased glomerular filtration rate, an index of renal function, in diabetic patients.

## 2. Methods

### 2.1. Study Participants

Data were obtained from the 2016–2017 Korean National Health and Nutrition Examination Survey (KNHANES), which was conducted by the Korea Centers for Disease Control and Prevention. The KNHANES is a self-reported survey administered to South Koreans of all ages and designed to gather annual national data on this population’s sociodemographic, economic, and health-related conditions and behaviors. Of the 16,277 survey participants, we excluded 15,790 participants who were aged <30 years because they were not subjected to blood screening tests conducted by the KNHANES (n = 4916), were not diagnosed with diabetes mellitus (n = 10,288), were diagnosed with kidney failure (n = 15), were restricted in daily life and social activities because of their health problems (n = 181), and were not representative of covariates considered in the study (n = 390). Accordingly, the final sample size included 487 (men: 274; women: 213) participants (Figure 1).

This study was an analysis of existing data; thus, it did not require approval from an ethics review board. The data used in this study were from the KNHANES, which has been annually reviewed and approved by the Korea Centers for Disease Control Research Ethics Review Committee since 2007.

### 2.2. Variables

In this study, the main dependent variable was renal function, which was determined by the Modification of Diet in Renal Disease (MDRD)-estimated glomerular filtration rate (eGFR). The glomerular filtration rate describes the flow rate of filtered fluid from the renal glomerular capillaries into Bowman’s capsule per unit time, and it is traditionally considered a credible index of renal function in health and disease [15]. The MDRD model, derived by an equation using serum creatinine, age, ethnicity, and sex, is recommended by many professional guidelines to assess renal function: MDRD eGFR = 175 × Serum Cr^–1.154^ × age^–0.203^ × 1.212 (if the patient is black) × 0.742 (if the patient is female) [16].

The main independent variable was the usage of food labels. The KNHANES contained the following question to be answered: “Do you read food labels when you buy food products?”; this usage of food labels was categorized as “Yes”, “No”, or “Unaware”, depending on the participants’ reports. “Yes” was defined as reading food labels when choosing food products to consume, “No” was defined as being aware of the food label but not being affected by it when choosing food products to consume, and “Unaware” was defined as being unaware of food labels.

Sociodemographic, economic, and health-related factors were also considered. Sociodemographic factors included the participant’s region of residence, educational level, household composition, and marital status. Economic factors included household income and current economic activity status. Health-related factors included the duration of diabetes, diabetes treatment, global recommendations on physical activity developed by the World Health Organization, solitary eating status, daily energy intake, smoking status, drinking status, obesity measured by the body mass index, hypertension, and menopausal status.

### 2.3. Statistical Analysis

Frequencies and mean eGFR were calculated for each of the categorized variables included in the study. Analysis of variance (ANOVA) was performed to compare the mean eGFR within each variable (Table 1). The presented p-value in Table 1 serves to indicate whether or not there is a significant difference in mean eGFR between the categorized groups. 

Multiple regression analysis was used to estimate the association between reading food labels and renal function after controlling for age, diabetes duration, diabetes treatment status, region, educational level, economic activity status, household income, household composition, marital status, physical activity, solitary eating status, energy intake, 24-h urine sodium excretion, smoking status, drinking status, BMI, hypertension status, and menopausal status (in female participants) (Table 2). In addition, multiple regression analysis of subgroups stratified by age, household composition, marital status, and solitary eating status was performed (Table 3). The main aim of the subgroup analysis is to identify either a consistency or larger difference among the different living conditions (i.e., number of family members, presence/absence of a spouse) of the participants.

For all data analyses, we used SAS version 9.4 (SAS Institute, Inc., Cary, NC, USA), and the significance level was set at *p* < 0.05. All statistics have been calculated using sample weights assigned to the study participants. The sample weights were constructed by KNHANES to represent the Korean population by accounting for the complex survey design and survey non-response.

## 3. Results

Table 1 presents the general characteristics of the diabetic patients who were included in the study sample (men: 274; women: 213). Both male and female diabetic patients who reported that they were unaware of food labels had mild loss of renal function compared to those who reported that they were aware of food labels. In 129 male diabetic patients who reported that they were unaware of food labels when choosing food products for consumption, the mean eGFR was 76.6 mL/min/1.73 m^2^. In 93 female diabetic patients who reported that they were unaware of food labels when choosing food products for consumption, the mean eGFR was 77.9 mL/min/1.73 m^2^.

Table 2 presents the multiple regression analysis of the factors associated with renal function. Unawareness of food labels when selecting food products for consumption was marginally associated with the loss of renal function in male diabetic patients (*β* = –10.01, standard error (SE) = 5.08, *p* = 0.0506). In female diabetic patients, the association between food label use and renal function was not statistically significant.

Table 3 presents the multiple regression analysis of the association between food label use and renal function stratified by age, household composition, marital status, and solitary eating status. A stronger association was found between unawareness of food labels and loss of renal function in male diabetic patients older than 70 years of age than in those younger than 70 years of age (*β* = –24.74, SE = 9.38, *p* = 0.0091). Among male diabetic patients who lived in a one-generational household, those who reported that they were unaware of food labels were more likely to have loss of renal function than those who reported that they were aware of food labels (*β* = –18.70, SE = 5.53, *p* = 0.0009). Among male diabetic patients who did not have a spouse, those who reported that they were unaware of food labels were more likely to have loss of renal function than those who reported that they were aware of food labels (*β* = –51.10, SE = 16.08, *p* = 0.0018). In addition, the association between unawareness of food labels and the loss of renal function in male diabetic patients who had been eating alone for the past year was statistically significant (*β* = –15.34, SE = 7.57, *p* = 0.0445).

## 4. Discussion

Maintaining an optimal diet is an important lifestyle strategy for managing renal function [17]. In previous studies resembling ours, a high diet quality (based on Dietary Guidelines for Americans score) showed an association between lower incidences of incident chronic kidney disease and decreased eGFR [18,19]. If we assume that reading food labels will lead to people eating healthier and having greater nutrition knowledge, then we would expect food label readers to have better renal conditions. Our study indicated that diabetic patients who were unaware of food labels had mild loss of renal function. Given that sodium is associated with insulin sensitivity and kidney hemodynamics, it is recommended that diabetic patients avoid sodium-rich foods and use food labels in order to identify food that is appropriate for managing their conditions [20]. This association has been explained as an effect of compensatory hyperinsulinemia sustaining renal tubular reabsorption of sodium [21]. High sodium intake leads to increased peripheral resistance and, therefore, could be responsible for loss of renal function and vascular relaxation [21,22]. This is relevant to our study’s findings. Our data showed the sodium intake of the sample diabetic patients using the 24-hour urine sodium excretion (mmol/L) provided by KNHANES. Diabetic patients with the highest quartile of 24-hour urine sodium excretion had the lowest mean eGFR compared to those with the lower quartiles of 24-hour urine sodium excretion (men: 77.5; women: 78.9).

The effect of food label unawareness on the probability of loss of renal function was statistically significant in only male diabetic patients. According to a previous study, women tend to be more nutritionally knowledgeable than men, and knowledge has no effect on food label use, implying that the existence of food labels may not affect women’s choice of food [23]. This finding seems relevant to the sample examined in our study because a relationship between food label unawareness and loss of renal function was not found in female diabetic participants.

Dietary practices are integral to overall health in older adults [24]. Additionally, dietary practices are complex, as they are determined by various individual, social, and environmental factors that may change over time. For instance, an older population suffers from deterioration in oral health, loss of physical ability needed to prepare meals, and a reduction in social engagement, all of which may contribute to poor nutrition-related attitudes [25]. Our study also suggested a stronger association between unawareness of food labels and renal function in male diabetic patients older than 70 years of age than in those younger than 70 years of age. Understanding psychosocial and physical changes that influence dietary behaviors in the older population could be substantial in future interventions to improve their quality of life.

Our study indicates that male diabetic patients of a one-generational household were more likely to have loss of renal function when they were unaware of food labels. Family is the primary source of caregiving for people, and food preparation is often a valued role within the family unit [26]. The effects of various living arrangements on health outcomes, particularly people living in a one-generational household, have been investigated in developed countries. Generally, the well-being of people living in a one-generational household is more vulnerable than that of those living in larger families [27,28]. Living arrangements could have a direct effect on dietary attitude. Those who live in a one-generational household have been proven to have a poor dietary habit because of their psychosocial circumstances, such as frequent solitary eating and less family care, all of which may exacerbate poor selection of food [29,30].

The same phenomenon was observed in male diabetic patients living without a spouse. Of the male diabetic patients who did not have a spouse, those who were unaware of food labels were more likely to have mild loss of renal function than those who read food labels. Studies have shown that women engage in healthier eating than men, specifically of fruits and vegetables, and this seems to be due to a better awareness of dietary value in women [31,32]. In the study community, it is a common social normal for women to be primarily responsible for meal preparation and food shopping, whereas men remain largely unfamiliar with food preparation [33,34]. Thus, being spouseless (e.g., unmarried, divorced, or widowed) indicates a barrier to consuming recommended foods for managing diabetes in male participants.

The act of eating alone has been a consistent nutritional risk associated with a higher risk of mortality [35,36,37]; however, the mechanisms underlying this association are still not fully understood, and information regarding how eating with others can help improve dietary practices has not been sufficiently discussed. Evidence suggested that a diet can be facilitated by commensality through normative functions. According to a previous study, sharing food with others has a positive impact on diet because a person’s nutritive requirements are usually understood by caregivers, family members, or friends [38]. By having a companion during mealtimes, a person is able to receive support and/or regulation for managing his/her condition. This could possibly explain the result in our study, which indicated an association of unawareness of food labels in conjunction with solitary eating with loss of renal function in male diabetic patients.

This study has several limitations. This study was a cross-sectional study and was unable to provide a causal relationship between food label usage and renal function. The KNHANES also uses self-report questionnaires. Thirdly, the data extracted may have been subject to recall bias. Furthermore, the presence or absence of medication was not considered in the study because the KNHANES did not contain survey questions regarding medicine prescription. It was unable to identify what kind of medications the participants were taking and their duration of use. Additionally, we were unable to identify other diabetes complications or diseases. Lastly, the sample size of our study was relatively small compared to previous studies. However, our study also has strengths. The dataset generated from the KNHANES is nationally representative of the health status of South Koreans. Additionally, the KNHANES is updated annually to incorporate the changes in real-life health circumstances of South Koreans. Finally, the KNHANES has been extremely useful in health-related studies, and it provides meaningful insights for South Korean health policies.

In conclusion, our study suggests that unawareness of food labels when choosing food products for consumption is associated with loss of renal function in the South Korean male diabetic population. Furthermore, the association was stronger among male diabetic patients who lived in a one-generational household, did not have a spouse, and ate alone. Further cohort and longitudinal studies are warranted to verify our findings and develop future interventions to manage the progression of renal disease in diabetic patients.

## Figures and Tables

**Figure 1 ijerph-17-01945-f001:**
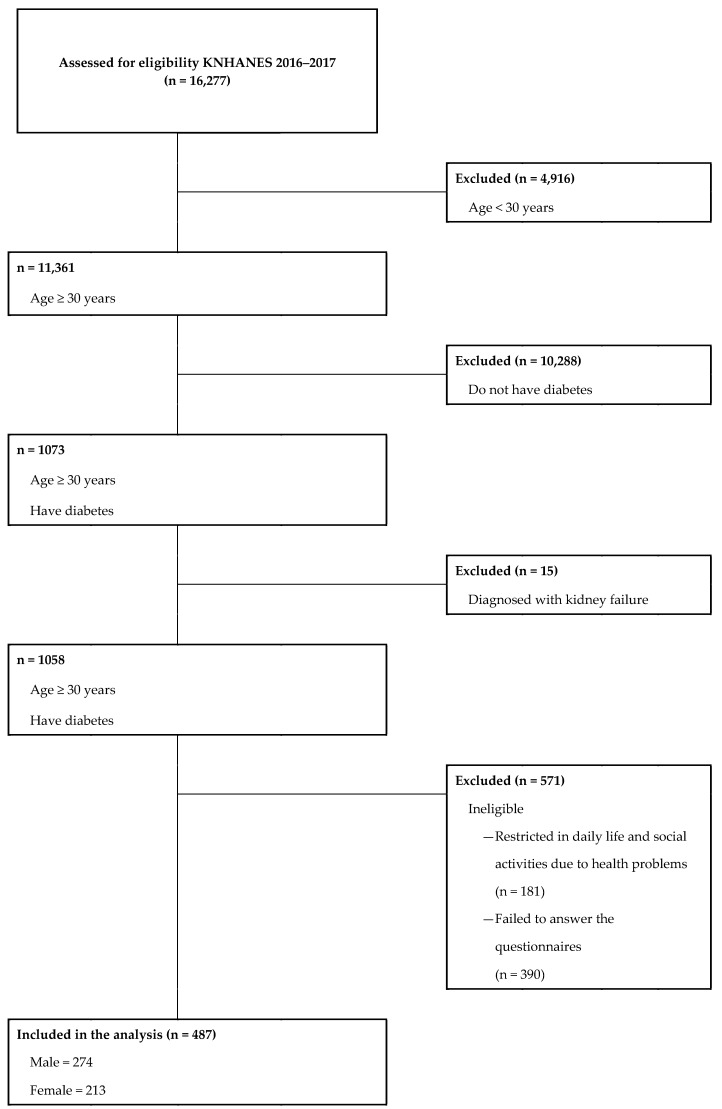
Flow diagram of subject inclusion and exclusion.

**Table 1 ijerph-17-01945-t001:** General characteristics of the study subjects.

Variables	Renal Function											
	eGFR (mL/min per 1.73 m2) ^†^											
	Total			Male					Female			
	N	%		N	%	Mean ± SD	*P*-Value		N	%	Mean ± SD	*P*-Value
**Food label usage**							0.0018					0.0012
Yes	62	12.7		18	6.6	89 ± 21.5			44	20.7	88.9 ± 19	
No	203	41.7		127	46.4	83 ± 17.2			76	35.7	86 ± 20.7	
**Unaware**	222	45.6		129	47.1	76.6 ± 19.5			93	43.7	77.9 ± 17.5	
Age							0.0002					0.0034
30–59	139	28.5		79	28.8	88.9 ± 17.2			60	28.2	92.7 ± 19.6	
60–69	166	34.1		90	32.8	79.4 ± 18.8			76	35.7	81.6 ± 20	
≥70	182	37.4		105	38.3	74.7 ± 18.2			77	36.2	77 ± 15.9	
**Diabetes duration (years)**							0.0019					0.1696
0–3	137	28.1		72	26.3	80.7 ± 15.4			65	30.5	84.9 ± 20.7	
4–6	97	19.9		54	19.7	81.4 ± 20.5			43	20.2	86.6 ± 16.6	
7–9	66	13.6		38	13.9	90.4 ± 19.8			28	13.1	87.5 ± 17.7	
≥10	187	38.4		110	40.1	76.1 ± 18.8			77	36.2	77.9 ± 19.7	
**Diabetes treatment (on-going)**							0.2696					0.1285
Yes	471	96.7		264	96.4	80.1 ± 19			207	97.2	82.6 ± 19.2	
No	16	3.3		10	3.6	85.6 ± 17.6			6	2.8	97.4 ± 24.9	
**Region**							0.886					0.5412
Metropolitans	224	46		121	44.2	80.3 ± 19.4			103	48.4	82.3 ± 19.4	
Rurals	263	54		153	55.8	80.4 ± 18.7			110	51.6	91 ± 18.5	
**Educational level**							0.1179					0.7634
≤Highschool	295	60.6		211	77	80.5 ± 19.3			84	39.4	88.9 ± 18.9	
≥College	192	39.4		63	23	79.9 ± 17.9			129	60.6	79.3 ± 19	
**Economic activity status**							0.4351					0.0174
Yes	249	51.1		165	60.2	83.1 ± 19			84	39.4	88.9 ± 18.9	
No	238	48.9		109	39.8	76.2 ± 18.2			129	60.6	79.3 ± 19	
**Household income**							0.4369					0.1915
Low	164	33.7		83	30.3	76.9 ±20.7			81	38	77 ± 17.3	
Mid–low	114	23.4		64	23.4	77.4 ± 18.1			50	23.5	84.9 ± 19.8	
Mid–high	111	22.8		67	24.5	80.8 ± 17.2			44	20.7	87.6 ± 15.6	
High	98	20.1		60	21.9	87.7 ± 17.4			38	17.8	88.3 ± 24.2	
**Household composition**							0.8035					0.633
One generational household	293	60.2		162	59.1	77.9 ± 18.3			131	61.5	80.5 ± 17.8	
≥ Two generational household	194	39.8		112	40.9	83.9 ± 19.5			82	38.5	87.1 ± 21.3	
**Marital status**							0.636					0.7511
Living w/ spouse	375	77		234	85.4	80.5 ± 18.6			141	66.2	84.9 ± 20.1	
Living w/o spouse	112	23		40	14.6	79.4 ± 21.1			72	33.8	79.4 ± 17.7	
**Physical activity**							0.0344					0.6334
Active	174	35.7		106	38.7	78.6 ± 15.4			68	31.9	85.2 ± 21.3	
Inactive	313	64.3		168	61.3	81.4 ± 20.9			145	68.1	82 ± 18.5	
**Solitary eating status**							0.3452					0.4701
Yes	153	31.4		69	25.2	77.8 ± 17.8			84	39.4	81.5 ± 20.7	
No	334	68.6		205	74.8	81.2 ± 19.3			129	60.6	84.1 ± 18.7	
**Energy intake (kcal)**							0.2568					0.2371
Quintile 1	96	19.7		54	19.7	79.3 ± 18.6			42	19.7	77.4 ± 16.8	
Quintile 2	98	20.1		55	20.1	80 ± 19.2			43	20.2	86.6 ± 18.4	
Quintile 3	99	20.3		56	20.4	77.7 ± 16.1			43	20.2	81.1 ± 22.4	
Quintile 4	97	19.9		54	19.7	77.1 ± 18.9			43	20.2	83.9 ± 19.8	
Quintile 5	97	19.9		55	20.1	87.5 ± 20.5			42	19.7	86.3 ± 18.9	
**24-h urine sodium excretion (mmol/L)**							0.5864					0.7004
Quartile 1	122	25.1		69	25.2	83.1 ± 19.3			53	24.9	84.8 ± 19.2	
Quartile 2	121	24.8		68	24.8	80.1 ± 21.2			53	24.9	81.4 ± 20.1	
Quartile 3	123	25.3		69	25.2	80.7 ±17.7			54	25.4	87.1 ± 20.3	
Quartile 4	121	24.8		68	24.8	77.5 ± 17.4			53	24.9	78.9 ± 17.7	
**Smoking status**							0.6152					0.13
Current smoker	93	19.1		85	31	83.4 ± 19.7			8	3.8	74.4 ± 28.8	
Non-smoker (never or former)	394	80.9		189	69	79 ± 18.5			205	96.2	83.4 ± 19	
**Drinking status**							0.9192					0.3382
2–4 times/week	125	25.7		108	39.4	82 ± 19.4			17	8	82.1 ± 19.2	
2–4 times/month	95	19.5		66	24.1	78.8 ± 17.5			29	13.6	81.4 ± 20.9	
Never or occasionally	267	54.8		100	36.5	79.5 ± 19.4			167	78.4	83.4 ± 19.4	
**BMI‡**							0.0216					0.6517
Obese (≥25)	234	48		125	45.6	78.4 ± 19.1			109	51.2	82.4 ± 20.7	
Normal or under-weight (<25)	253	52		149	54.4	82 ± 18.8			104	48.8	83.7 ± 18.2	
**Hypertension**							0.7417					0.4183
Hypertension	314	64.5		173	63.1	79.2 ± 19.7			141	66.2	81.9 ± 19	
Prehypertension	88	18.1		53	19.3	81.5 ± 17.6			35	16.4	82.7 ± 19.7	
Normal	85	17.5		48	17.5	83.4 ± 17.5			37	17.4	87.9 ± 21	
**Menopause**												0.1858
Yes									199	93.4	81.9 ± 19.2	
No									14	6.6	99.9 ± 16	
**Year**							0.0828					0.1802
2016	236	48.5		131	47.8	81.3 ± 18.9			105	49.3	81.5 ± 21.4	
2017	251	51.5		143	52.2	79.5 ± 19			108	50.7	84.6 ± 17.4	
**Total**	487	100		274	56.3	79 ± 17.7			213	43.7	81.1 ± 19.6	

BMI: body mass index; † Modification of Diet in Renal Disease estimated glomerular filtration rate (mL/min per 1.73 m^2^) = 175 × (SCr)^–1.154^ x (age)^–0.203^ × 0.742 (if female); ‡ Obesity status defined by BMI based on 2014 Clinical Practice Guidelines for Overweight and Obesity in Korea.

**Table 2 ijerph-17-01945-t002:** Multiple regression analysis for renal function.

Variables.	Renal Function																						
	eGFR (mL/min per 1.73 m^2^) ^†^																						
	Model 1								Model 2								Model 3						
	Male				Female				Male				Female				Male				Female		
	β	SE	*P*-Value		β	SE	*P*-Value		β	SE	*P*-Value		β	SE	*P*-Value		β	SE	*P*-Value		β	SE	*P*-Value
**Food label usage**																							
Yes	Ref.				Ref.				Ref.				Ref.				Ref.				Ref.		
No	–4.5	4.78	0.348		–2.53	5.8	0.6636		–7.33	4.78	0.1268		–6.78	7.31	0.3555		–5.49	4.87	0.2616		–3.21	4.69	0.4948
Unaware	–7.51	4.67	0.1099		–1.93	5.92	0.7453		–12.77	4.89	0.0098		–11.89	6.66	0.0764		–10.01	5.08	0.0506		–0.3	5.14	0.9528
**Age**																							
30–59	Ref.				Ref.												Ref.				Ref.		
60–69	–8.08	2.96	0.0071		–12.28	6.14	0.0475										–5.65	3.09	0.0694		–10.46	5.08	0.0415
≥70	–2.11	3.49	0.0007		–17.74	7.8	0.0246										–9.32	3.64	0.0113		-15.63	6.43	0.0163
**Diabetes duration (years)**																							
0–3	Ref.				Ref.				Ref.				Ref.				Ref.				Ref.		
4–6	3.93	2.8	0.1623		0.73	5.91	0.9017		2.02	3.19	0.527		–4.94	5.89	0.4032		1.97	2.95	0.5047		0.35	4.07	0.9318
7–9	10.81	3.81	0.0052		–2.83	7.62	0.7104		8.26	3.44	0.0174		–7.34	7.83	0.3505		9.08	3.86	0.0198		–1.68	5.15	0.7453
≥10	2.3	2.61	0.3804		–3.88	6.67	0.5613		-1.32	2.69	0.6242		–8.25	6.07	0.1759		0.9	2.67	0.7353		-3	4.31	0.4872
**Diabetes treatment (on-going)**																							
Yes	Ref.				Ref.				Ref.				Ref.				Ref.				Ref.		
No	8.31	4.53	0.0687		1.65	20.6	0.9364		9.23	4.87	0.0597		2.52	18.43	0.8916		7.16	3.99	0.0747		4.22	13.2	0.7499
**Region**																							
Metropolitans	Ref.				Ref.												Ref.				Ref.		
Rurals	0.71	2.53	0.7791		1.04	4.39	0.8132										–0.12	2.46	0.962		0.7	2.63	0.7891
**Educational level**																							
≤Highschool	4.35	2.21	0.0506		–0.84	6.83	0.9024										3.56	2.3	0.1224		1.1	4.65	0.8135
≥College	Ref.				Ref.												Ref.				Ref.		
**Economic activity status**																							
Yes	Ref.				Ref.												Ref.				Ref.		
No	–1.15	2.42	0.634		–12.11	4.68	0.0107										–1.96	2.5	0.4344		–8.42	3.34	0.0128
**Household income**																							
Low	-4.78	3	0.1128		–3.88	7.58	0.61										–4.42	3.41	0.1968		–2.69	5.28	0.6106
Mid–low	-3.69	3.71	0.3224		5.08	8.47	0.5492										–2.33	4.1	0.5708		4.76	5.63	0.3991
Mid–high	–2.2	3.18	0.4902		11.72	7.81	0.136										–2.31	3.84	0.5489		11.71	5.25	0.0274
High	Ref.				Ref.												Ref.				Ref.		
**Household composition**																							
One generational household	0.8	2.45	0.7457		0.68	5.15	0.8945										0.3	2.48	0.9051		1.75	3.8	0.646
≥Two generational household	Ref.				Ref.												Ref.				Ref.		
**Marital status**																							
Living w/ spouse	Ref.				Ref.												Ref.				Ref.		
Living w/o spouse	2.78	3.12	0.3734		3.91	5.22	0.4549										5.68	3.56	0.1127		1.39	5.71	0.8074
**Physical activity**																							
Active									Ref.				Ref.				Ref.				Ref.		
Inactive									3.75	2.29	0.1041		–2.33	4.93	0.6376		4.3	2.23	0.0553		0.8	3.37	0.8121
**Solitary eating status**																							
Yes									Ref.				Ref.				Ref.				Ref.		
No									–4.93	2.31	0.0345		–2.06	4.92	0.6754		–4.63	2.69	0.087		5.24	5.01	0.2977
**Energy intake**																							
Quintile 1									–0.42	3.04	0.8898		–7.14	9.12	0.4347		0.01	3.08	0.9979		–9.74	5.55	0.0813
Quintile 2									–1.23	3.3	0.7101		3.74	8.74	0.6692		–0.39	3.2	0.9037		1.52	5.99	0.8
Quintile 3									Ref.				Ref.				Ref.				Ref.		
Quintile 4									–0.92	3.01	0.7604		–3.71	7.73	0.6322		–0.39	2.92	0.8939		–1.13	5.3	0.8307
Quintile 5									5.99	3.71	0.1083		2.08	9.39	0.8254		4.67	3.94	0.2382		0.01	5.38	0.9979
**24-h urine sodium excretion (mmol/L)**																							
Quartile 1									Ref.				Ref.				Ref.				Ref.		
Quartile 2									–0.25	3.04	0.9349		–7.37	6.29	0.243		–0.36	3.04	0.9063		–0.71	4.42	0.8729
Quartile 3									–0.83	3.4	0.808		4.28	7.69	0.5785		1.27	3.67	0.7307		7.84	5.59	0.1628
Quartile 4									–1.87	3.38	0.5813		–3.05	6.32	0.6305		–0.99	3.41	0.7713		5.23	5.27	0.3224
Smoking status																							
Current smoker									4.07	2.69	0.1326		1.48	13.42	0.9125		2.4	2.73	0.3798		–2.22	9.74	0.8199
**Non-smoker (never or former)**									Ref.				Ref.				Ref.				Ref.		
Drinking status																							
2–4 times/week									1.14	2.55	0.6557		4.92	8.73	0.574		0.42	2.75	0.8802		4.05	5.86	0.491
2–4 times/month									–0.96	2.84	0.7352		1.45	7.89	0.8543		–0.25	2.81	0.9284		–1.94	5.55	0.7268
Never or occasionally									Ref.				Ref.				Ref.				Ref.		
**BMI‡**																							
Obese (≥25)									–4.85	2.6	0.0639		–2.69	5.08	0.5978		–4.9	2.61	0.0623		–1.86	3.39	0.5834
Normal or under-weight (<25)									Ref.				Ref.				Ref.				Ref.		
**Hypertension**																							
Hypertension									–4.84	3.03	0.1116		–6.31	6.96	0.3663		–3.57	3.06	0.2453		1.53	4.51	0.7349
Prehypertension									–3.14	3.65	0.3913		–3.93	9.38	0.676		–2.85	3.42	0.4064		1.39	5.86	0.813
Normal									Ref.				Ref.				Ref.				Ref.		
**Menopause**																							
Yes													–15.45	8.12	0.0592						–6.94	6.37	0.2776
No													Ref.								Ref.		
**Year**																							
2016	2.23	2.04	0.2751		–6.6	4.77	0.1686		4.67	2.38	0.0516		–5.04	4.38	0.2517		3.94	2.33	0.0934		–4.79	3.21	0.1388
2017	Ref.				Ref.				Ref.				Ref.				Ref.				Ref.		

Model 1: Odds ratio after controlling for duration of diabetes, diabetes treatment status, sociodemographic variables, and economic variables; Model 2: Odds ratio after controlling for duration of diabetes, diabetes treatment status, health-related variables, and nutrition-related variables; Model 3: Odds ratio after controlling for all variables included in the study; BMI, body mass index; ^†^ Modification of Diet in Renal Disease estimated glomerular filtration rate (mL/min per 1.73 m^2^); eGFR = 175 × (SCr)^–1.154^ × (age)^–0.203^ × 0.742 (if female); ^‡^ Obesity status defined by BMI based on 2014 Clinical Practice Guidelines for Overweight and Obesity in Korea.

**Table 3 ijerph-17-01945-t003:** Subgroup analysis stratified by age, household composition, marital status, and solitary eating status.

Variables	Renal Function†								
	Food Label Usage								
	Yes		No				Unaware		
	β		β	SE	*P*-Value		β	SE	*P*-Value
**Male**									
**Age**									
30–59	Ref.		–8.02	5.82	0.1704		–2.94	7.18	0.6824
60–69	Ref.		–3.2	8.3	0.7		–12.14	8.29	0.1452
≥70	Ref.		–14.87	9.18	0.1073		–24.74	9.38	0.0091
**Household composition**									
One generational household	Ref.		–7.42	5.34	0.1663		–18.7	5.53	0.0009
≥ Two generational household	Ref.		–6.79	7.26	0.3508		–5.2	7.84	0.5084
**Marital status**									
Living w/ spouse	Ref.		–5.45	5.57	0.3287		–9	5.84	0.1251
Living w/o spouse	Ref.		-35.16	13.82	0.0119		–51.1	16.08	0.0018
Solitary eating status									
Yes	Ref.		–10.5	7.49	0.1629		–15.34	7.57	0.0445
No	Ref.		–5.09	5.98	0.3958		–9.89	6.31	0.119
**Female**									
**Age**									
30–59	Ref.		7.65	6.63	0.2505		8.21	11.18	0.4639
60–69	Ref.		7.65	6.63	0.2505		8.21	11.18	0.4639
≥70	Ref.		5.5	9.12	0.5474		6.52	8.3	0.4336
**Household composition**									
One generational household	Ref.		–1.88	4.99	0.7067		–3.14	4.76	0.5108
≥Two generational household	Ref.		2.73	6.17	0.6585		12.2	7.99	0.129
**Marital status**									
Living w/ spouse	Ref.		2.02	4.9	0.6815		2.49	5.06	0.6237
Living w/o spouse	Ref.		–2.83	8.49	0.739		0.47	9.79	0.9621
**Solitary eating status**									
Yes	Ref.		–3.14	9.05	0.7291		–4.05	9.07	0.6561
No	Ref.		–7.06	4.24	0.0985		–3.29	5.11	0.5209

†Modification of Diet in Renal Disease estimated glomerular filtration rate (mL/min per 1.73 m^2^); eGFR = 175 × (SCr)^–1.154^ × (age)^–0.203^ × 0.742 (if female).

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
