# Peer review of "Association between Food Label Unawareness and Loss of Renal Function in Diabetes: A Cross-Sectional Study in South Korea"

_ijerph, 2020, doi:10.3390/ijerph17061945_

Round 1

Reviewer 1 Report

'Sex differences in the association between food label unawareness and loss of renal function in diabetes: A cross-sectional study in South Korea' is a well-written manuscript on a topic that is important to prevention and clinical intervention of patients with diabetes. My main concerns lie with the statistical analyses and conceptualization of the paper. 

  1. Though the title mentions sex differences, there is no development of literature addressing sex differences known on the topics in the introduction. Given the current analyses, it would also be helpful to discuss other demographic differences that are known.
  2. It would also be useful to see hypotheses or research questions following the purpose statement. 
  3. Under section 2.2, paragraph 2, what was the question stem to assess usage of food labels?
  4. The statistical analyses need to be more clearly described either in the statistical analyses section or in the description of the results. For example, what does adjusting for covariates mean? which variables used a t-test?
  5. there is no discussion of the assumptions of regression.
  6. need to add a discussion of power to statistical analyses section.
  7. In results, what does 'weighted' age mean?
  8. Were sex differences examined as preliminary analyses in relation to other demographic variables? 
  9. In table 1, it is not clear what test was used to determine the p values?
  10. Table 1 would be easier to read if a vertical line is added to the right of the total %.
  11. the number of male participants who were 'aware' is quite small, but this is not addressed in analyses or discussion of limitations. 
  12. what is meant by 'stratified by age...' in table 3. where is the justification to stratify by these variables? 
  13. the regressions in tables 2 and 3 do not appear to be sufficiently powered, but it is not entirely clear. Instead of the way the data are currently analyzed, I recommend adding a 'covariate section' where demographic differences are reported for eGFR and food label awareness. and then follow up with a regression predicting eGFR only entering the variables that were significant in the model. You could then check for sex differences through interaction/moderation or by running separate models if there is sufficient power. 
  14. If the focus on the paper is on sex differences..., then the introduction, analyses and discussion all need to be centered on this theme. Perhaps the title is misleading and should be revised.
  15. check for minor grammatical errors and awkwardly written sentences throughout. 

Author Response

We were pleased to have an opportunity to revise our paper. In revising the paper, we have carefully considered your comments and suggestions. As instructed, we have attempted to succinctly explain changes made in reaction to all comments. Comments were very helpful overall, and we are appreciated of such constructive feedback on our original submission. After addressing the issues raised, we feel the quality of the paper is much improved and hope you agree. Our response to each comment follows, and we attached a revision note and also underline the revised sections of the manuscript. Again, thank you for the valuable and helpful comments.

The point-by-point reponse is attached as WORD file.

Reviewer 2 Report

Thanks for your manuscript for review process.

This manuscript need to be revised several points.

First of all, there is no description of the number.

Reviewers do not know which sentences to point out.     1) What is the hypothesis for this study? There is a description of the background. However, the author needs to specify the hypothesis.   2)The author has described only the P value in Table 1.
But please also provide the critical values.
    3) Authors descrived that "This study has several limitations. First, this study was a cross-sectional study and was unable to provide a causal relationship between food label usage and renal function. Second, the KNHANES uses self-report questionnaires, and third, the data extracted may have been subject to recall bias. However, our study also has several strengths. The dataset generated from the KNHANES is nationally representative of the health status of South Koreans. Additionally, the KNHANES is updated annually to incorporate the changes in real-life health circumstances of South Koreans. Finally, the KNHANES has been extremely useful in health-related studies, and it provides meaningful insights for South Korean health policies." as a limitations.  However, I think there are many other limitations in this research. For example, the presence or absence of medication, the presence or absence of a disease. The author needs to be more specific.    

Author Response

We were pleased to have an opportunity to revise our paper. In revising the paper, we have carefully considered your comments and suggestions. As instructed, we have attempted to succinctly explain changes made in reaction to all comments. Comments were very helpful overall, and we are appreciated of such constructive feedback on our original submission. After addressing the issues raised, we feel the quality of the paper is much improved and hope you agree. Our response to each comment follows, and we attached a revision note and also underline the revised sections of the manuscript. Again, thank you for the valuable and helpful comments.

The point-by-point response is attached as a WORD file.

Reviewer 3 Report

This paper is overall well designed, conducted, discussed and well-referenced. I have listed two suggested changes below.

Suggested changes:

  1. Results. Line 2, change "diabetes usually develops at an old age" to: "type 2 diabetes usually develops at an older age".                     Reason: This is not true for type 1 diabetes. Also, "older" is a less pejorative term.
  2. Also, for the western world, assumptions such as " It is common social norm for women to be primarily responsible for meal preparation.. may still exist in some cultures but they are no longer the norm. I suggest adding qualifiers, such as, " in the study community" or, "in South Korea", etc.

Author Response

(The authors gave the same response as above.)

Round 2

Reviewer 2 Report

Thanks for this revision. This manuscript has revised based on the reviewers coments. So, this manuscript will be able to publish.